# Usher Syndrome in the Inner Ear: Etiologies and Advances in Gene Therapy

**DOI:** 10.3390/ijms22083910

**Published:** 2021-04-10

**Authors:** Evan M. de Joya, Brett M. Colbert, Pei-Ciao Tang, Byron L. Lam, Jun Yang, Susan H. Blanton, Derek M. Dykxhoorn, Xuezhong Liu

**Affiliations:** 1Department of Otolaryngology, University of Miami Miller School of Medicine, Miami, FL 33136, USA; edejoya@med.miami.edu (E.M.J.); bmc48@miami.edu (B.M.C.); peictang@med.miami.edu (P.-C.T.); sblanton@med.miami.edu (S.H.B.); 2Dr. John T. Macdonald Foundation Department of Human Genetics, John P. Hussman Institute for Human Genomics, University of Miami Miller School of Medicine, Miami, FL 33136, USA; ddykxhoorn@med.miami.edu; 3Medical Scientist Training Program, University of Miami Miller School of Medicine, Miami, FL 33136, USA; 4Bascom Palmer Eye Institute, University of Miami School of Medicine, Miami, FL 33136, USA; blam@med.miami.edu; 5John A. Moran Eye Center, Department of Ophthalmology and Visual Sciences, University of Utah, Salt Lake City, UT 84132, USA; jun.yang@hsc.utah.edu; 6Interdisciplinary Stem Cell Institute, University of Miami Miller School of Medicine, Miami, FL 33136, USA

**Keywords:** syndromic hearing loss, retinitis pigmentosa, CRISPR, antisense oligonucleotides

## Abstract

Hearing loss is the most common sensory disorder with ~466 million people worldwide affected, representing about 5% of the population. A substantial portion of hearing loss is genetic. Hearing loss can either be non-syndromic, if hearing loss is the only clinical manifestation, or syndromic, if the hearing loss is accompanied by a collage of other clinical manifestations. Usher syndrome is a syndromic form of genetic hearing loss that is accompanied by impaired vision associated with retinitis pigmentosa and, in many cases, vestibular dysfunction. It is the most common cause of deaf-blindness. Currently cochlear implantation or hearing aids are the only treatments for Usher-related hearing loss. However, gene therapy has shown promise in treating Usher-related retinitis pigmentosa. Here we review how the etiologies of Usher-related hearing loss make it a good candidate for gene therapy and discuss how various forms of gene therapy could be applied to Usher-related hearing loss.

## 1. Introduction

### Hearing Loss

The ear is a highly evolved sensory organ with a complex mechanotransduction pathway. As such, there is significant room for genetic changes leading to hearing loss [1]. The ear can be divided into three anatomical sections—the outer ear (from the visible pinna to the ear canal), the middle ear (from the tympanic membrane to the oval window including the smallest bones in the body), and the inner ear (the semicircular canal, vestibule, and cochlea). Hearing begins when sound is funneled into the ear canal by the pinna [2]. The waves then travel down the canal and come in contact with the tympanic membrane, also known as the ear drum. Here, sound waves are translated into mechanical vibrations and are passed down the bones of the middle ear, from the malleus to the incus to the stapes and through the oval window into the cochlea. The cochlea is a spiral-shaped structure filled with potassium-rich endolymph. Vibrations propagate through the endolymph and deflect the stereocilia of hair cells in the organ of Corti. Movements of the stereocilia result in the opening of ion channels on the hair cells, producing an action potential. Through this process mechanical sound waves are converted into electrical impulses that travels down the vestibulocochlear nerve (CNVIII) to be processed by the brain. Failure at any one of these steps will result in hearing reduction or loss [1,3,4].

Hearing loss (HL) is the most common sensory deficit. Estimates place the prevalence of disabling hearing loss at around 466 million people worldwide representing about 5% of the population [5]. Approximately half of all cases of severe to profound hearing loss have a genetic basis [6,7]. HL is often the result of a defect in a single gene which can be inherited in an autosomal dominant, autosomal recessive, X-linked, or mitochondrial manner. Hearing loss is classified as syndromic if there are other clinical manifestations in addition to the hearing loss, and non-syndromic if hearing loss is the only symptom of the genetic variant [8,9,10]. Non-syndromic HL makes up eighty percent of cases of genetic hearing loss. Among the non-syndromic forms of HL, 60–75% of these are inherited in an autosomal recessive manner, 20–30% are autosomal dominant, with a small minority (~2%) resulting from X-linked or mitochondrial inheritance [11,12]. To date, there have been more than 150 genes and 6000 causative variants implicated in genetic causes of hearing loss [12,13,14,15,16]. The remaining ~20% of genetic hearing loss cases are syndromic.

An example of syndromic hearing loss is Usher syndrome. Usher syndrome is characterized by hearing loss, visual impairment due to retinitis pigmentosa (RP), and in some cases, vestibular dysfunction [17]. Usher syndrome is the most common cause of deaf-blindness and an important etiology underlying autosomal recessive deafness.

While some of the genes resulting in Usher syndrome have been identified for some time there has been limited progress in therapeutic options for hearing loss in these patients [18]. Supportive techniques such as hearing aids or cochlear implants are the only options for those subtypes that are amenable [19]. The advent of genome editing technologies, such as clustered regularly interspaced short palindromic repeats (CRISPR)/Cas9, has brought the promise of gene therapies within reach. In this review, we focus on the progress towards novel therapies for Usher syndrome and how these can be applied to Usher-related hearing loss.

## 2. Usher Syndrome

### 2.1. Classification of Usher Syndrome

Clinically, Usher syndrome is differentiated into three distinct types—Usher syndrome type 1 (USH1), Usher syndrome type 2 (USH2), and Usher syndrome type 3 (USH3). These types are differentiated by the degree of hearing impairment, age of visual problem onset, and the presence or absence of vestibular dysfunction [17,20].

Usher type 1 has the most severe manifestations and accounts for 25 to 44% of all Usher patients [21]. USH1 patients present with severe-to-profound bilateral sensorineural hearing loss (SNHL) and RP at an average age of nine years [22,23]. If hearing loss is left untreated, speech development may be interrupted [23]. Vestibular areflexia is present in all patients with USH1 and leads to balance deficits with children experiencing delayed walking and motor development [24].

Usher type 2 patients have moderate-to-severe congenital HL, which is less pronounced than the HL in USH1 [25,26]. USH2 is the most common type, accounting for more than 50% of all observed cases [21,27]. USH2 patients retain normal vestibular function and experience a later onset of RP in the second decade of life with an average age of 17 years [22,28].

Usher type 3 is the least common type, accounting for an estimated 2 to 5% of all Usher cases [27,29]. Hearing loss in USH3 progressively worsens as individuals age [30]. RP onset in USH3 is variable, but typical onset is around age 20 years [24]. USH3 patient also experience vestibular phenotypes. In all three types of Usher syndrome, hearing loss occurs before the onset of visual impairment [31]. Balance deficits in USH1 and USH3 can be aggravated further by the loss of vision due to RP [18].

### 2.2. Genetics of Usher Syndrome

The relatively low frequency of Usher syndrome and heterogeneity in populations makes it difficult to understand its true prevalence. Usher syndrome is a significant cause of autosomal recessive hearing loss among deaf and hard-of-hearing children. Estimates of prevalence vary by country, ranging from 3 to 6.2 per 100,000, but up to 16.7 per 100,000 in American children [29,32,33].

Within each type of Usher syndrome there are numerous subtypes associated with various genes and loci (Table 1). At least 11 causative or suspected genes have been identified, six associated with USH1, three with USH2, and two with USH3. Digenic inheritance in Usher syndrome—mutations in two Usher-related genes–has been proposed, though the existence of digenic inheritance is contradicted by recent analyses [31]. Additional loci have been mapped, such as USH1E, USH1H, USH1K, and USH2B, but the underlying gene has not been identified [34,35,36,37].

Due to regional founder effects, Usher subtype composition varies by population. Among French Canadians in Quebec, USH1C is the major genetic USH1 subtype due to a founder mutation [64], while the R245 * founder mutation associated with USH1F was found to be a significant cause of USH1 in Ashkenazi Jewish populations [65].

USH3 is the least common of all Usher types, though it is found more frequently in Ashkenazi Jewish and Finnish populations [65,66,67,68]. In many of these cases, founder mutations account for the increased prevalence of a specific Usher syndrome type. For example, all Finnish patients with USH3A carry the same mutation Y176 * associated with USH3A [66].

### 2.3. Usher Protein Function

Proteins encoded by Usher genes have various roles in the development and function of the inner ear (Table 1). Usher proteins are most abundant in the stereocilia and the synaptic regions of hair cells and guide the cohesion and development of hair bundles [27]. Mysoin VIIa (USH1B), harmonin b (USH1C), and cadherin 23 (USH1D) are responsible for the cohesion of the stereocilia, the shaping of the hair bundle and cadherin 23 and protocadherin 15 are integral to the structure of the tip links [69]. The SANS protein (USH1G) is a scaffolding protein that associates with harmonin b protein [49,70]. ADGRV1 (USH2C) and usherin (USH2A) are components of ankle links, filaments connected to the base of stereocilia [55,71]. Further, WHRN (USH2D) has been observed to interact with ADGRV1 and usherin at the ankle links and co-localize with these proteins at the synaptic regions of outer hair cells in the cochlea to promote synaptic adhesion and development [72]. CLRN-1 (USH3A) is a hair bundle protein with four transmembrane domains necessary for the shape and development of the bundle and ribbon playing a role in mechano-electrical transduction [61,73,74].

## 3. Progress and Challenges in Gene Therapy for Usher Syndrome

### 3.1. Current Treatment Modalities for Usher-Related Hearing Loss

Current therapies for HL range from sound amplification to cochlear implantation (CI). The current treatment for USH1- and USH3-induced hearing loss is CI [23,75]. Hearing loss caused by USH2 is typically less severe, and the current intervention is hearing aids [76]. The existing interventions for Usher-related hearing loss are invasive and expensive, and gene therapy presents an opportunity to greatly improve outcomes through restoration of an individual’s endogenous hearing, ending the reliance on devices that do not faithfully recapitulate normal hearing [77,78,79]. Hearing aids are limited in providing high frequency access, which is imperative for speech understanding. With cochlear implants, while they improve audibility up to the normal range, the signal is highly degraded. These limitations for invasive and non-invasive treatments do not allow for full restoration of hearing resulting in poorer speech understanding than normal hearing peers, specifically in situations with significant background noise.

Early diagnosis and intervention in Usher syndrome are critical to improving prognosis [80]. Particularly in USH1, sign language is a decreasingly effective method of communication due to the early onset of RP and subsequent vision loss [81]. Currently, Usher syndrome is diagnosed at a mean age of 10, with USH1 diagnosed earlier than USH2 or USH3 [23]. Achieving earlier diagnosis of Usher syndrome is a requisite for the successful development and delivery of gene therapy for the inner ear.

### 3.2. Therapy for Usher-Related Retinitis Pigmentosa

Although gene therapy for RP caused by any etiology has shown promise, in this review we will focus on treatments for retinitis pigmentosa caused by Usher pathogenic genes. Using ClinicalTrials.gov (Search terms: Condition or Disease–“retinitis pigmentosa”, other terms–“Usher AND therapy”) relevant trials were identified. Results were screened to include only trials targeting RP associated with an Usher subtype using gene transfer, editing, or suppression therapy.

Our search identified two relevant trials (Table 2). In the SAR421869 trial, an equine infectious anemia-virus-based (EIAV) lentiviral vector expressing MYO7A was delivered via subretinal injections to patients with USH1B. The Phase I/IIA trial was terminated by the company for a review of clinical development plans and priorities after the license was returned from Sanofi to Oxford Biomedica [82]. In the QR-421a trial, an antisense oligonucleotide is being used to target a defective exon 13 in USH2A patients. The trial, currently in Phase I/II is ongoing, and results have not yet been released [83].

### 3.3. Defining Gene Therapy

Gene therapy presents a promising potential treatment especially for monogenic diseases. Gene therapies include the insertion, deletion, or replacement of whole genes, or the editing of existing pathogenic sequences within the endogenous copies of the gene. For each intended effect, there are different therapeutic modalities. Gene replacement modalities utilize a vector, typically a virus, to deliver a functional copy of a defective gene. Gene editing modalities, such as the CRISPR/Cas9 system, are able to modify the DNA sequence by inducing double strand breaks and subsequent homology-directed repair using a template DNA containing the sequence of interest. This precision allows for the introduction of the specific DNA variant while leaving the remaining sequence unaltered.

### 3.4. Gene Therapy for the Inner Ear

The inner ear is an attractive target for gene therapy. Attempts to correct Usher-related HL thus far has mainly focused on mouse models. The only gene therapy targeting the inner ear to reach human clinical trials is CGF166, a recombinant adenovirus 5 (Ad5) vector containing cDNA encoding ATOH1 (Hath1) delivered via intra-labyrinthine infusion [86]. This study was designed to assess the safety of intra-labyrinthine infusion as well as changes in pure tone audiometry compared to pre-treatment levels over the course of two years. The trial, sponsored by Novartis, was suspended in late 2019 due to an apparent lack of efficacy. The trial only enrolled individuals with profound hearing loss, which may have had an effect on the outcome.

### 3.5. Challenges in Gene Therapy

Usher gene protein products have been implicated in the development of two key structures in the inner ear—the hair bundles and ribbon synapse [87,88]. While these structures present a clear target for future therapies, hearing loss in USH1 and USH2 is congenital. At the time of birth, the damage has already been done; human hair cells are fully mature with response to sound at 25–27 weeks gestation [89]. In mice, the inner ear is still maturing at birth, and auditory function does not occur until about postnatal day 15. This temporal difference in murine and human inner ear development means that a successful intervention in a mouse model may not directly translate to humans. Applied to humans, post-natal therapy in a mouse model would need to take place in utero, and the implications of these trials are limited unless the therapy is delivered to a mature inner ear [90].

## 4. Applying Gene Therapy Modalities to Usher-Related Hearing Loss

### 4.1. Gene Therapy: Gene Modulation and Editing

#### 4.1.1. Antisense Oligonucleotides and RNAi

RNA interference (RNAi) technology uses small RNA molecules to guide the sequence-specific silencing of a target gene by binding to the complementary sequence and inducing mRNA degradation. This approach has been shown to effectively target therapeutically relevant genes, particularly those with variants inherited in an autosomal dominant, gain-of-function manner [91]. The use of RNAi as a therapeutic approach for Usher syndrome will be far more limited since it is an autosomal recessive disease resulting from defective or absent proteins. Antisense oligonucleotides (ASO) are short sequences of RNA/DNA that can bind to complementary RNA strands, inhibiting translation. ASO therapy has been used primarily in mouse models of USH1C, specifically to prevent the transcription of a *c.216 G > A* mutation which creates an aberrant splice site resulting in a frameshift and premature stop. Various studies have shown both auditory and vestibular rescue (Table 3) [92].

#### 4.1.2. Genomic Editing

The CRISPR/Cas9 system emerged in 2013, as a novel technology for genome editing [101]. The technology harnesses the base sequence structure of DNA to make precise changes at selected nucleotides on the genome. CRISPR (clustered regularly interspaced short palindromic repeats) serves as an “adaptive immune” system for bacteria allowing them to recognize and remove pathogenic bacteriophage sequences and prove the integrity of their genomes [102]. CRISPR involves the use of guide RNA sequences which bear complementarity to the target DNA sequence flanked directly by a protospacer-adjacent motifs (PAMs) an essential sequence for the functionality of the CRISPR system [103]. This RNA–DNA base pairing flanked by the PAM sequence defines the CRISPR targeting specificity facilitating the recruitment of the Cas9 endonuclease that induces a double strand DNA break. This double strand break is corrected by the cell through either non-homologous end joining (NHEJ) or homology-directed repair (HDR). The most commonly used Cas9 is derived from *Streptococcus pyogenes* but Cas9 or Cas9-related proteins with variable PAM sequences have been identified in a variety of bacteria and archaea. The fixing of the double strand DNA breaks using non-homologous end joining is an imprecise process that leads to the introduction of insertions and deletion around the cut site. On the other hand, providing a template DNA strand to guide the homology-directed repair allows for the precise repair of the break and provides an opportunity to alter the DNA sequence in a controlled manner (e.g., the introduction of a single nucleotide variant in the DNA sequence). Prior to the advent of CRISPR/Cas9, zinc-finger-nuclease-induced double-stranded breaks was explored as a method of gene repair of an USH1C p.R31 * mutation [104]. Although CRISPR/Cas9-based genome editing has been used extensively in the laboratory, the transition to clinical application is a more difficult undertaking. Fuster-Garcia et al. had some success targeting the c.229delG mutation in fibroblasts from a patient with USH2A with an RNA-Cas9 ribonucleoprotein and homologous recombination repair with an engineered template [105]. The CRISPR/Cas9 has been used in the murine inner ear; Gao et al. demonstrated the successful use of a Cas9-guide RNA complex in a mouse model of autosomal dominant deafness [91]. To date, there have not been inner ear animal models studying the efficacy of gene editing technology in Usher syndrome.

### 4.2. Gene Therapy: Gene Replacement

#### 4.2.1. Viral Delivery

When selecting a viral vector, many factors must be considered. The vector cargo capacity must be able to accommodate the length of the replacement gene, and the vector must effectively transduce the target cell type within the body. The immunoreactivity of the virus is also an important factor, with an ideal vector eliciting little to no immune response.

#### 4.2.2. Types of Viruses Used in Gene Therapy

Different viral vectors have been explored in the inner ear including adenovirus, adeno-associated virus (AAV), retrovirus, HSV, and lentivirus [106,107,108]. Since they do not integrate into the host genome, adenovirus vectors are unable to confer long-term expression of the target gene. However, compared to adenoviruses, adeno-associated viruses (AAVs) have emerged as a good candidate vector due to their decreased immune response and ability to confer sustained expression of the gene product in the inner ear [109].

#### 4.2.3. Viral Vectors for the Inner Ear (Anc80L65)

Though immunoreactivity to AAV has been observed to be low, as an endemic virus, prevalence of pre-existing immunity to AAV approaches 50% in some populations [110]. Anc80L65, the ancestor of AAV serotypes 1, 2, 8, and 9, was identified by Zinn et al. through in silico analysis as an ideal candidate due to its ability to circumvent existing immunity to better deliver genes to the inner ear [110]. Furthermore, in a mouse model of USH1C (c.216 G > A), the AAV2/Anc80L65 achieved stabilization of hair cells and bundles, reducing the degeneration and preserving function [93]. The Anc80 virus was found to outperform all other AAVs in level of expression in inner and outer hair cells in a mouse model with stable expression at one month follow-up. In addition to expression in the cochlea, the virus also showed expression in human vestibular tissue [109].

Other viruses have been explored; a synthetic AAV2.7m8 has been used successfully in the inner ear to target cochlear hair and supporting cells, and AAV9-PHP.B rescued hearing loss in Usher 3A genes (CLRN-1) [98,111]. AAVs in the inner ear are known to more effectively transduce inner hair cells (IHCs) than outer hair cells (OHCs), with AAV2 and AAV8 providing the most effective delivery [112,113]. All animal-modeled therapies for Usher syndrome in the literature are outlined in Table 3.

A vector must accommodate the sequence within its own genome and deliver it to the target tissue. Viral capacity varies; the AAV can carry a transgene of up to 4 kb, while a dual-AAV system can be deployed to deliver larger payloads (6kb) to the inner ear [114]. Usher pathogenic target genes vary in length and location, ranging from 561 base pairs (bp) to 18,918 bp with between three and 90 exons (Table 4).

#### 4.2.4. Gene Stabilization

There is minimal research on gene stabilization methods. In USH3A, the CLRN1 c.144T > G, p.Asn48Lys missense mutation results in unstable gene products. A small molecule, BioFocus 844, was identified through successive screenings, and mitigated the progression of USH3A in a Tg;KI/KI mouse model through the stabilization of the dysfunctional protein product [115].

#### 4.2.5. Exon Skipping

Often in a large protein, such as many of the Usher-related proteins, there is only a small mutation that leads to the loss of function of the whole protein. These large proteins also often contain subunit redundancy. It has thus been recognized that there is potential to skip any exons that contain a mutation and still end up with a functional or partially-function protein. Skipping has been carried out by use of small RNAs, antisense oligonucleotides (ASO), and CRISPR. Exon skipping has had some success in addressing diseases such as Duchenne’s muscular dystrophy [116,117], and has potential for use in Usher syndrome.

The deep intronic variant c.7595-2144A > G in *USH2A* causes a splice donor site in intron 40 which in turn causes the addition of a 152 bp pseudoexon into the transcript. This leads to a premature stop and a non-functional protein [118]. Slijkerman et al. were able to use an ASO to target the pseudoexon and effectively skip it to restore the expression of wild type *USH2A* in effected fibroblasts [119]. This proved to be an effective option to evaluate clinically for c.7595-2144A > G *USH2A* mutations, however this is a relatively rare mutation. Additionally, any therapy involving ASOs would require appropriate delivery vehicles and repeat dosing.

The most common *USH2A* mutation is c.2299delG [120]. Pendse et al. targeted this mutation in a mouse model harboring a mutation in *Ush2a* exon 12, which is analogous to the human exon 13. They employed CRISPR/Cas9 editing to knock out and skip exon 12 in the mice. The resultant product properly localized in the cochlea and was able to restore hair cell structure and auditory function in the mice [121].

### 4.3. Delivery Challenges

One obstacle in the progression from molecular therapy to clinical use is efficient delivery of the bioactive molecules to the correct target cells. The complexity of the bony-labyrinthine barrier presents a challenge for drug delivery, but its unique structure also creates opportunity for novel approaches [122]. Previously, the inner ear was believed to be immune privileged due to the lack of lymphatic drainage from the cochlea, though macrophages are now known to migrate into and reside in the cochlea [123]. This may make the cochlea more amenable to viral delivery with reduced potential for immune engagement. The immune response to viral delivery platforms is complex, and it is difficult to determine the long-term safety of transgene expression [122]. Multiple strategies to reduce the immune response to AAV viral vectors have been proposed, including selection of AAV-naïve subjects, use of highly efficient AAVs and engineered AAV serotypes, or the administration of immunosuppressant drugs in combination with the viral vector delivery [124].

Physical delivery of these therapeutics also poses its own set of challenges. The cochlea is set in the temporal bone, and as such must be accessed surgically [125]. There are several techniques to deliver these potential therapeutics to the inner ear including round window membrane and oval window membrane injections [126], canalostomy with injection in the posterior semicircular canal [97], and utricle injection [127]. The most common and promising are methods that support the diffusion of therapeutics through the cochlea are round window membrane and oval window membrane injections such as round window injection.

## 5. Discussion

### 5.1. From Animal Models to Human Trials

While the majority of in vivo research has been conducted in mouse models, Gyorgy et al. found that AAV9-PHP.B mediated almost complete transduction to OHCs and IHCs in a non-human primate model [98]. Jüttner et al. found that in AAV-targeting glial cells in the retina, the probability of an AAV targeting the same cell class was 0.32, compared to a 0.67 probability between non-human primates and humans. As a result, vectors optimized to target cell types in murine models may not directly translate to human use. The study also noted, however, that the probability of an AAV expressing in a given cell group in humans if it does not express in mice is low, meaning that mice may be a useful model for screening and restricting the number of AAVs to be further tested in humans and non-human primates [128].

Ultimately, progression to clinical trials in humans relies on the translation of animal model results to human physiology. The trail by Gyorgy et al. is the only trial of a gene therapy targeting an Usher etiology that has included a non-human primate [98]. Though there remain numerous cultural and practical barriers to studies in non-human primates, these trials ease the translation from animal models to clinically viable therapy [129].

### 5.2. Overcoming Vector Capacity

Some Usher-related genes are very large and beyond the capacity of a single viral vector. The carrying capacity of the viral vectors can be overcome through implementing dual and triple recombinant vectors. Akil et al. demonstrated the successful delivery of otoferlin (DFNB9) to the murine inner ear [130]. Dual and triple AAV modalities should be considered for treatment of Usher subtypes with pathogenic genes larger than single AAV capacity of 4.7 kb (*MYO7A, PCDH15, ADGRV1, CDH23*, and *USH2A*).

Exon skipping is also a promising method for compensation of loss of function when the whole gene product is too large to be efficiently delivered to the target cells [119,121].

### 5.3. Prevention vs. Correction

The previously described therapies have been delivered to the developing murine inner ear, as early postnatal mice have inner ears that are still developing. In the current treatment setting, diagnosis in humans is largely made following the development of a fully formed cochlea and dysfunctional hair cells. Therapies deployed before or during development are valuable, but future trials must consider the therapeutic modalities targeting the developed inner ear.

Earlier diagnosis of Usher patients would aid in early intervention. With up to 90% of Usher patients having disease caused by mutations in known USH genes, increasing molecular diagnosis efforts and identifying specific genetic etiologies is an important step in the therapeutic process [131].

### 5.4. Conclusions

Usher syndrome is the leading cause of deaf-blindness. The genetic etiologies have been extensively studied. Recent progress using gene-based therapies for Usher-related retinitis pigmentosa gives hope that similar techniques can be applied to the inner ear to ameliorate the deleterious effects of Usher mutations on human hearing.

## Figures and Tables

**Table 1 ijms-22-03910-t001:** Usher type, subtype, and associated genes.

Type	Subtype	Gene Involved	Protein Product	Protein Function	NSHL Classification	References
Usher Type 1	USH1B	*MYO7A*	Myosin VIIA	Actin-binding molecular motor	DFNB2/ DNFA11	[38,39,40]
	USH1C	*USH1C*	Harmonin	Scaffolding protein	DFNB18	[41,42,43]
	USH1D	*CDH23*	Cadherin 23	Cell adhesion molecule (Upper tip-link)	DFNB12	[44,45]
	USH1F	*PCDH15*	Protocadherin 15	Cell adhesion molecule (Lower tip-link)	DFNB23	[46,47,48]
	USH1G	*USH1G*	Sans	Scaffolding protein	-	[49]
	USH1J	*CIB2 **	Calcium and integrin binding protein	Calcium and integrin binding protein	DFNB48	[50,51,52,53]
Usher Type 2	USH2A	*USH2A*	Usherin	Cell adhesion molecule	-	[54,55]
	USH2C	*ADGRV1 ***	Adhesion G-coupled receptor V1	Adhesion G-coupled receptor V1	-	[56]
	USH2D	*WHRN*	Whirlin	Scaffolding protein	DFNB31	[57,58,59]
Usher Type 3	USH3A	*CLRN1*	Clarin-1	Transmembrane protein used in scaffolding and cellular trafficking	-	[60,61,62]
	USH3B	*HARS ****	Histidine tRNA ligase	Histidine tRNA ligase	-	[63]

* CIB2 involvement in Usher is contested. ** Formerly GPR98 or VLGR1. *** Research on *HARS* in Usher is limited. NSHL = non-syndromic hearing loss; classification numbers were included if mutations also lead to NSHL.

**Table 2 ijms-22-03910-t002:** Gene therapy for retinitis pigmentosa.

Therapeutic	Usher Subtype	Gene Involved	Phase Reached	Status
UshStat (SAR421869) [84]	USH1B	*MYO7A*	Phase I/IIA	Suspended, 2017 Terminated, 2019
QR-421a [85]	USH2A	*USH2A*	Phase I/II	Ongoing

**Table 3 ijms-22-03910-t003:** Overview of animal-modeled therapies for Usher syndrome in the inner ear.

Type	Subtype	Mutation	Vector/Molecule	Delivery Method	Intervention Time	Hair Cell Penetrance	Auditory Outcome	Vestibular Outcome	Source
Type 1	USH1C	*c.216 G > A*	AAV2/Anc80L65	RWM injection	P0–P1		Auditory rescue at low frequencies	Rescued	Pan [93]
	USH1C	*c.216 G > A*	Antisense oligonucleotide	Intraperitoneal injection	P3–P5		Rescued, comparable to WT	Rescued, comparable to WT	Lentz [94]
	USH1C	*c.216 G > A*	Antisense oligonucleotide	Intraperitoneal injection	P10		Partially rescued	Rescued, comparable to WT	Lentz [94]
	USH1C	*c.216 G > A*	Antisense oligonucleotide	Amniotic microinjection	E13–E13.5		Partially rescued	Rescued, comparable to WT	Depreux [95]
	USH1C	*c.216 G > A*	Antisense oligonucleotide	Intraperitoneal injection	P5				Depreux [95]
	USH1C	*c.216 G > A*	Antisense oligonucleotide	Transuterine microinjection in the otocyst	E12.5		Rescued	Rescued	Wang [92]
	USH1G	*Ush1g ^−/−^*	AAV8	RWM injection	P2.5	IHC: 87% ± 4% (apex), 45% ± 6% (base) IHC: 33% ± 6% (apex), 25% ± 5% (base)	Partial restoration	Rescued	Emptoz [96]
Type 2	USH2D	*Ush2a Whirler*	AAV8-whirlin	Posterior semicircular canal injection	P4	IHC:77.1% ± 12.7% OHC:13.3% ± 9.82%	Most hearing improvement at 8 kHz	Rescued	Isgrig [97]
Type 3	USH3A	*Clrn1^−/−^*	AAV9-PHP.B	RWM injection	P1	IHC:60–80% OHC:30–40%	Rescued hearing at low frequencies (4–8 kHz)		György [98]
	USH3A	*Clrn1^ex4fl/fl^ Myo15-Cre^+/−^*	AAV2/8-Clrn1	RWM injection	P1–P3	IHC:90% OHC:20%	Restored hearing		Dulon [99]
	USH3A	*Clrn1^−/−^ KO TgAC1*	AAV2-Clrn1-UTR AAV8-Clrn1-UTR	RWM injection	P1–P3		Partially restored (20–30 dB shy of WT)		Geng [100]

**Table 4 ijms-22-03910-t004:** Table of Usher gene size, exons, and location in the genome (Used hg38 in the UCSC (University of California Santa Cruz) Databank).

Gene	Total Exons	Base Pairs	Location
*MYO7A*	49	86,996	11q13.5
*USH1C*	27	50,517	11p15.1
*CDH23*	70	419,028	10q22.1
*PCDH15*	38	998,461	10q21.1
*USH1G*	3	7173	17q25.1
*CIB2*	6	26,840	15q25.1
*USH2A*	73	800,553	1q41
*ADGRV1*	90	605,641	5q14.3
*WHRN*	12	103,394	9q32
*CLRN1*	4	44,773	3q25.1

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
