# Peer review of "Usher Syndrome in the Inner Ear: Etiologies and Advances in Gene Therapy"

_ijms, 2021, doi:10.3390/ijms22083910_

Round 1
Reviewer 1 Report
This is an interesting review about etiologies and advances in Gene therapy in Usher syndrome.
However, I have some remarks to improve this review.
- First, there are some mistakes in the Table 3 (that the authors must be renamed as Table 4, because the authors mentioned Table 4 (line 316) before Table 3 (line 321) in the text).
USH1C has 27 exons, CDH23 70 exons, USH2A 72 exons. For PCDH15, there are three main isoforms CD1, CD2 and CD3, that represents a total of 37 exons. For CLRN1, the longest isoform has 4 exons. Base pairs values are false for these genes.
- The authors did not mention the Y-linked mode of inheritance (DFNY1 and DFNY2).
- The authors could be abbreviating Usher syndrome as USH all along the manuscript.
- They use the abbreviation RP for retinitis pigmentosa, but they did not use it in the paragraph “Therapy for Usher-related Retinits Pigmentosa” line 176-191.
- In the “Usher protein function” part, line 136-151, the authors did not mention the tip-link of which the major constituents are CDH23 and PCDH15.
- Table 1: indicate that MYO7A is also a DFNA gene (DFNA11). Remove the question mark after USH1G; the gene name is USH1G.The authors must used Uniprot to check the name of each protein.
- Use the HGVS recommendation to write the variant (* or Ter and not X for nonsense variant).
- Line 134: USH3A not USH3
- Line 328: c.144T>G;p.(Asn48Lys)
- Line 408: Add USH2A and CDH23 (see the modification in Table 3/4).
- In the References part, the references 18 and 33 are the same, like 79 and 82; 95 and 97.
- Check the reference 55. The reference is about USH2 patients but not USH1 or USH3? Is there another publication?
- I am not sure about the pertinence of some references. For example, the authors sometimes referred to the original article as reference 17 (1969) to describe Usher syndrome but they used a reference of 2015 (ref 20) to talk about the three types of USH. The description of the differences between Usher subtypes is much earlier than 2015. Check all the references used.
- Line 161: there is a problem with the format bibliography. Indicate [57-59] instead of [57],[58-59].
- Line 426: I think the authors can mention another reference with a number of patients much higher than 67 patients.
Author Response
Author Responses are in Red
This is an interesting review about etiologies and advances in Gene therapy in Usher syndrome.
However, I have some remarks to improve this review.
- First, there are some mistakes in the Table 3 (that the authors must be renamed as Table 4, because the authors mentioned Table 4 (line 316) before Table 3 (line 321) in the text). This has been fixed
USH1C has 27 exons, CDH23 70 exons, USH2A 72 exons. For PCDH15, there are three main isoforms CD1, CD2 and CD3, that represents a total of 37 exons. For CLRN1, the longest isoform has 4 exons. Base pairs values are false for these genes. Thank you for pointing the issue. We have fixed it.
- The authors did not mention the Y-linked mode of inheritance (DFNY1 and DFNY2). We focused on etiologies presented in the hereditary hearing loss home page (https://hereditaryhearingloss.org)
- The authors could be abbreviating Usher syndrome as USH all along the manuscript. Thank you. We decided to leave it as is to ensure consistency in the manuscript
- They use the abbreviation RP for retinitis pigmentosa, but they did not use it in the paragraph “Therapy for Usher-related Retinits Pigmentosa” line 176-191. Abbreviation was added
- In the “Usher protein function” part, line 136-151, the authors did not mention the tip-link of which the major constituents are CDH23 and PCDH15. This has been added
- Table 1: indicate that MYO7A is also a DFNA gene (DFNA11). Remove the question mark after USH1G; the gene name is USH1G.The authors must used Uniprot to check the name of each protein. Table has been updated
- Use the HGVS recommendation to write the variant (* or Ter and not X for nonsense variant). Thank you for pointing this out, these have been fixed
- Line 134: USH3A not USH3 This has been fixed
- Line 328: c.144T>G;p.(Asn48Lys) this has been added
- Line 408: Add USH2A and CDH23 (see the modification in Table 3/4). This has been added
- In the References part, the references 18 and 33 are the same, like 79 and 82; 95 and 97. These have been reconciled
- Check the reference 55. The reference is about USH2 patients but not USH1 or USH3? Is there another publication? We have checked the reference 55, which is about CIB2. It is correct.
- I am not sure about the pertinence of some references. For example, the authors sometimes referred to the original article as reference 17 (1969) to describe Usher syndrome but they used a reference of 2015 (ref 20) to talk about the three types of USH. The description of the differences between Usher subtypes is much earlier than 2015. Check all the references used. Thank you, the earlier papers have been added and cited where appropriate. The more recent manuscripts remain as an updated review of the topic
- Line 161: there is a problem with the format bibliography. Indicate [57-59] instead of [57],[58-59]. This has been remedied
- Line 426: I think the authors can mention another reference with a number of patients much higher than 67 patients. We have checked the reference and feel it is appropriate. We would appreciate if there is more specific suggestion on the reference that we could refer to.
Reviewer 2 Report
This is an informative and well-written review focused on gene-based therapies for HL in Usher syndrome. I have the following minor comments:
- Line 69: An instead of “a” example
- Table 1: NSHL is not defined
- Table 1: what do you mean by “causes NSHL”?
- I would move up section 4.1 up so that the definition is provided before getting to details.
- Line 379: “approach is” seems redundant. Sentence is not grammatically correct as written.
- Line 420-424: paragraph needs correction and maybe more explanation about why further molecular research studies are needed. What do you mean by “both sides”?
- Table 4: are the studies all from mouse model? Please specify.
Author Response
Author responses are in Red
This is an informative and well-written review focused on gene-based therapies for HL in Usher syndrome. I have the following minor comments:
- Line 69: An instead of “a” example thank you, this has been fixed
- Table 1: NSHL is not defined this has been defined
- Table 1: what do you mean by “causes NSHL”? table has been updated in the foot notes to clarify the meaning
- I would move up section 4.1 up so that the definition is provided before getting to details. The section has been moved
- Line 379: “approach is” seems redundant. Sentence is not grammatically correct as written. This has been fixed
- Line 420-424: paragraph needs correction and maybe more explanation about why further molecular research studies are needed. What do you mean by “both sides”? this section has been updated to be more clear
- Table 4: are the studies all from mouse model? Please specify. Yes, all cited studies were done in the mouse model. We have specified in the line 248